# Multiband Photogrammetry and Hybrid Image Analysis for the Investigation of a Wall Painting by Paolo de San Leocadio and Francesco Pagano in the Cathedral of Valencia

**DOI:** 10.3390/s23042301

**Published:** 2023-02-18

**Authors:** Max Rahrig, Miguel Ángel Herrero Cortell, José Luis Lerma

**Affiliations:** 1Photogrammetry and Laser Scanning Research Group (GIFLE), Department of Cartographic Engineering, Geodesy and Photogrammetry, Universitat Politècnica de València, 46022 Valencia, Spain; 2Department of Audiovisual Communication, Faculty of Fine Arts, Documentation and History of Art, Universitat Politècnica de València, 46022 Valencia, Spain

**Keywords:** renaissance wall painting, non-destructive testing (NDT), non-invasive, cultural heritage, multispectral imaging (MSI), multiband imaging (MBI), photogrammetry, Cathedral of Valencia

## Abstract

A workflow for the photogrammetric combination of non-invasive multispectral imaging techniques ranging from ultraviolet (UV) and visible (VIS) to near infrared (NIR) for the investigation of wall paintings is presented. Hereby, different methods for image analysis and visualisation techniques are discussed. This includes the combination of spectral bands in hybrid false-colour images and image analysis by applying NDVI/NDPI and PCA. The aim of the research is to generate a high-resolution photogrammetric image set, providing information on underdrawings, material differences, damages, painting techniques and conservation measures. The image data are superimposed with pixel accuracy in a geographic information system (GIS) for further analysis, tracing of observations and findings and the annotation of further information. The research is carried out on the ‘Adoration of the Shepherds’, an early Spanish Renaissance wall painting created in 1472 by Paolo de San Leocadio and Francesco Pagano in the Cathedral of Valencia. The wall painting is preserved in an unfinished condition, and half of it is represented by the initial plaster and preparation layers. This gives the possibility to compare and evaluate the observations of the finished areas as well as carry out an in-depth study of the working techniques.

## 1. Introduction

In this study, a workflow for the photogrammetric combination of non-invasive and non-destructive multiband imaging (MBI) techniques for the investigation of wall paintings is presented. The aim of the research was to provide a set of high-resolution orthoimages superimposed in pixel-to-sub-pixel accuracy to detect material differences, damages, working techniques and conservation measures. Therefore, MBI techniques were used, covering the spectral ranges of long-wave ultraviolet (UV; 315–400 nm), visible light (VIS; 400–700 nm) and near infrared (NIR; 700–1000 nm). The different image data built the basis for the investigation of mural paintings. For this, principal component analysis (PCA) and an adapted variant of the normalized difference vegetation index (NDVI), called herein normalized difference painting index (NDPI) approach were used, as well as a combination of hybrid false-colour images. All image data were superimposed in a geographic information system (GIS) ready for further analysis, tracing of observations and findings and the annotation of further information. The research was carried out on the ‘Adoration of the Shepherds’, an early Spanish renaissance wall painting created in 1472 by Paolo de San Leocadio and Francesco Pagano in the Cathedral of Valencia.

The next paragraphs give some general information on the techniques and terminology used in this study, followed by a brief history and introduction of the object. Detailed information on the sensors, data acquisition and processing follows in Section 2 “Materials and Methods”, the final orthoimages and their interpretation and results are presented in Section 3 “Results”, followed by a discussion of the workflow (Section 4). The article ends with a brief conclusion (Section 5).

MBI, as the main technique used in this study, is a method of spectral imaging widely used and well established in the investigation of cultural heritage (CH). Hereby, different wavelengths or spectra of electromagnetic radiation are captured in image files representing spectral bands. However, the terminology of MBI is still not established properly. This is due to the inconsistencies in the general terminology of spectral imaging. Already the definitions of the two most known methods, multispectral imaging (MSI) and hyperspectral imaging (HSI), vary [1]. The main differentiation is given by the number of spectral bands captured, where MSI applies between four and ten bands, while in HSI up to several hundred bands are used [2,3]. In addition, the way the single image bands are captured is a common way to distinguish between HSI and MSI. For MSI, each spectral band is recorded separately by using different filters or detectors, while in HSI the entire spectral range is acquired in a single image file, called an image cube [4,5,6,7]. Even the bandwidth is used to distinguish between HSI and MSI. In HSI, the bandwidth is very narrow, around 10 nm or less, whereas in MSI it is around 10 to 40 nm [2,8,9]. However, the term MSI is also commonly used for spectral imaging with longer bandwidths, especially for the observation of CH [3,10]. For a better differentiation of spectral imaging between defined (narrow) band filters of up to 40 nm (MSI) and the use of broadband filters, several new terms have been introduced during the last few years. Cosentino prefers the term technical photography (TP) [11], while Keller et al. [12,13,14] and Herrero-Cortell et al. [9], as well as this study, follow the suggestions made by Picollo et al. [1] to use the term multiband imaging (MBI).

MBI is mostly done by using modified digital cameras, where the low-pass filter, which blocks infrared (IR) radiation, is removed or replaced by a protection glass, allowing the entire spectral range to pass through. This provides the possibility to use the entire spectral sensitivity of the image sensor. In the case of CMOS sensors, the sensitivity reaches from around 350 to 1100 nm. To capture only a certain spectral range, e.g., UV, VIS or NIR, specific filters are placed in front of the lens [10,15,16,17]. When the wavelength of an excitation source (light source) is also controlled, characteristic features of individual materials can be identified. Hereby two main ways of excitations are used in CH. The first is called reflectance, where the excitation is carried out in the same spectral range as the filters in front of the camera. The second is luminescence, where the excitation source provides less energetic light, in a lower spectral range, than the filters in front of the camera let pass through. A combination of reflectance and luminescence images can be used as a non-contact and non-destructive examination method in CH research to determine materials, e.g., coatings, damages and pigments used in a painting [10,15,17]. Table 1 gives an overview of combinations of induced and detected spectral ranges commonly used for MBI [10,12,13,15,18,19,20].

The abbreviations for the different imaging methods of MBI are inconsistent too. In 1968/69, van Asperen de Boer [21,22] introduced the application of Vidicon tubes to capture infrared reflectography images in the spectral range of short-wave infrared (SWIR, 1000–2500 nm). To distinguish between the reflected infrared photography of the NIR range (carried out with IR-sensitive film established in 1930) [23,24] and the new method, he used the abbreviation IR for reflected infrared photography and introduced IRR for infrared reflectography in the SWIR range. However, at least with the application of infrared luminescence imaging (IRF, IRL or VIL) [10,15,25,26,27], a heterogeneous terminology is mandatory to avoid mixing up between the spectral region of infrared (IR) and the spectral imaging method. Keller et al. introduced the most exact terminology [12,13,14,18], whereby first the detected spectral range is mentioned (UV; VIS; IR), then the imaging method, reflectance or luminescence imaging (R or L), followed by the spectral range of the radiation source (UV; VIS; IR): e.g., UV-R__UV_ stands for UV-induced UV-reflectography (c.f., Table 1). This study follows the terminology introduced by Keller. To distinguish between the two IR long-pass filters used, an extension to indicate the filter is added, e.g., IR-R__IR_850_ stands for IR-induced IR reflectography carried out with a long-pass filter starting at 850 nm.

Van Asperen de Boer already mentioned the importance of high-resolution imaging techniques to detect details and information in an adequate way to fit the needs of art-historical research and conservation science [22]. The spatial resolution of a digital image can be described by the ground sampling distance (GSD). The GSD describes the object distance of two-pixel centres of the image sensor. To visualise a detail in a digital image, the GSD needs to be three times smaller than the detail itself [28]. Due to limited sensor resolutions and the size and shape of the heritage object, it is often necessary to capture multiple images to cover the entire object in a proper GSD. For the combination, different techniques are used. For flat surfaces, image mosaicking is an adequate option to combine several images to capture a bigger area [19,29,30,31,32,33], but for complex geometries and to guarantee a correct scaling of the object, a photogrammetric workflow is mandatory. There are already several studies of MSI and MBI for the 3D documentation of heritage objects. Webb et al. demonstrated the quality of the 3D surface reconstruction between an unmodified camera and a modified camera by using either the VIS or the NIR spectral images, which provided comparable results [17]. Mathys et al. reconstructed the object’s surface by using the VIS-R__VIS_ images and processed the MSI texture mapping by replacing the colour images with the single spectral band images to generate 16 different MSI texture models [34]. Zainuddin et al. used a Micasense RedEdge multispectral camera with five different spectral images for the 3D spectral documentation of rock art [35]. Nocerino et al. mapped colour images and UV-induced visible luminescence images via a photogrammetric workflow on the 3D surface captured with a structured light scanner [36]. Pamart et al. introduced a mechanical structure with a multisensory camera platform to move a spectral imaging system at a constant distance in front of a wall painting to capture photogrammetric multiband image sets [37].

In this study, a more flexible approach to capturing and analysing high-resolution photogrammetric MBI data of wall paintings is presented. Furthermore, the application of hybrid images by combining different spectral bands into false colour images is presented, and the advantages of PCA and NDPI for the investigation of mural paintings discussed.

## 2. Materials and Methods

In this section, the methodology of the multiband image investigation is described. Therefore, the case study, the ‘Adoration of the Shepherds’ (Figure 1), an early Spanish renaissance wall painting created 1472 by Paolo de San Leocadio and Francesco Pagano in the Cathedral of Valencia, is presented first. This is followed by descriptions of the equipment of the different spectral imaging techniques, the data acquisition and an overview of the data processing, image analysis, hybrid image processing and further investigation in a geographic information system (GIS).

### 2.1. The Case Study

The present paper deals with one of the most iconic Spanish Renaissance paintings. It was possibly one of the very first attempts of frescoes with Italian methodologies in Spain, and, for sure, it is one of the most renowned examples, due to its own history [38].

During the celebrations of the Pentecost fest in the Cathedral of Valencia on 21 May 1469, a burning cotton torch unleashed a virulent fire, which devastated the silver altarpiece that crowned the main altar and the set of wall paintings that had decorated the main altar since 1432. The following year, a new pictorial project was undertaken for the reconstruction of the presbytery, which, however, would never be accomplished because its creator Nicolló Delli (1413–1470) died. The following year another frustrated project followed, this time by Pere Rexach and Anthoni Canyiçar, who were fired due to the poor quality of their results. Upon learning of the matter, Cardinal Rodrigo de Borja (later Pope Alexander VI) considered hiring two painters he trusted to travel to Valencia to take charge of the pictorial project. This is how he commissioned the Neapolitan Francesco Pagano, (who had worked in the Vatican rooms) together with a young painter from Reggio Emilia, Paolo de San Leocadio (trained in the environment of the artists who worked in the court of Ferrara). Thus, at the dawn of the summer of 1472, these two painters disembarked in the port of Valencia, ready to undertake the new pictorial project [39].

As had happened with Delli, barely two years before, the painters had to take a test in the internal enclosure of the Cathedral to prove that they truly knew and had mastered the technique of fresco painting, which was required of them. The case study that concerns us is the technical test or exam with the theme of the ‘Adoration of the Shepherds’ with which the Italian painters had to convince the Chapter of the Cathedral of their technical worth, skills and pictorial quality, demonstrating what they were able to achieve. The cathedral chapter and Cardinal Rodrigo de Borgia (later Pope Alexander VI) were immediately convinced of their skills, so the two artists had to start the altar paintings without even finishing the sample work. Thus, this case study stands directly at the very beginning of the Spanish Renaissance period, as the two artists are the founders of fresco painting in the Italian tradition in Spain, so this could constitute one of the very first pieces of evidence of Quattrocento fresco methodologies [40].

Due to the rediscovery of the vault paintings in the altar in 2004, much research has been done in recent years on Paolo de San Leocadio and Francesco Pagano and their work. The ‘Adoration of the Shepherds’ has, with a few exceptions, only been dealt with in passing. The focus has mostly been on the large vault paintings or the exciting story of the search for the right artists to execute the ceiling painting. Nevertheless, the mural impresses with its high-quality execution and offers special insights into the working methods of the two artists, especially due to its unfinished state, an especially useful condition in terms of technical artistic research [41].

Unlike the rest of the previous pictorial projects, we did not keep records of the pictorial materials used [42]. It was only verified in the Libre d’obres (‘Book of works of the Cathedral)’ that the walls where the test was going to be painted were pealed, the scaffolds were erected, and the corresponding proportions of lime and sand were bought. A new sieve was also acquiesced as well as large papers to prepare the cartons to transfer the drawing, but, unlike the previous projects, nothing was said on the pigments nor on the technique and methods used [40]. Thus, a good part of the interest in the study of this painting lies in the possibility of being able to register the painting at different points of the spectrum through image techniques. This is intended to collect evidence on the pictorial procedures and materials used by these painters.

During the last restoration measure, the painting was detached from the wall and prepared as a *strappo*. After its restoration, it is now presented in the entrance area to the Chapel of the Holy Chalice.

### 2.2. Data Acquisition

For a correct alignment of all spectral bands used in this study, and to guarantee a high-resolution metrical documentation of the mural painting, a photogrammetric workflow was followed. Therefore, several multiband images being UV-induced UV reflectography (UV-R__UV_), visible-induced visible reflectography (VIS-R__VIS_) and IR-induced IR reflectography (IR-R__IR_) were acquired. Additionally, a terrestrial laser scanning (TLS) survey was carried out.

The following equipment was used for the data acquisition:76 multiband images were acquired by using a Fujifilm IS Pro digital single lens reflect (DSLR) camera (Figure 2a), modified ex-stock as ultraviolet (UV), visible (VIS) and near-infrared (NIR) sensitive. Hereby sets of 19 images of UV-R__UV_, VIS-R__VIS_ and two IR-R__IR_ were taken;17 wide-angle images were taken using a Canon EOS 5D Mark II DSLR (Figure 2b). These images form the basic photogrammetric survey supporting the orientation of the MBI data;3 terrestrial laser scans serving as ground truth measurements for scaling and correct orientation of the image data were used. The TLS data were acquired by using a Trimble TX6 (Figure 2c).

#### 2.2.1. Multiband Image Acquisition

MBI was carried out using a Fujifilm IS Pro DSLR (Table 2) in combination with a CostalOptic UV-VIS-IR 60 mm 1:4 apochromatic macro lens. The DSLR was ex-stock, modified as a UV-VIS-IR sensitive camera. Therefore, the camera has no IR-blocking glass in the low-pass filter in front of the image sensor, as is usually done in digital cameras. This opens the camera sensitivity to the entire spectral range of the image sensor (350 to 1050 nm). With the adaption of special filters in front of the lens, it is possible to reduce the incoming light to a certain spectral band [9,10,15,17]. In this way UV-R__UV_, VIS-R__VIS_ and IR-R__IR_ images were taken. The filters and their spectral ranges are listed together with the other equipment used for MBI in Table 2. For the illumination of the VIS and IR images, two halogen studio lamps were used and two UV LEDs for the UV images. The UV LEDs provide mainly light around 365 nm, but also some parasitic light in the blue (around 400 nm) and in the red band (above 600 nm). Therefore, it was not possible to acquire ultraviolet-induced visible luminescence images (VIS-L__UV_) even if some tests on partial parts of the painting were carried out with a filtered UV lamp. However, since images were not taken of the whole painting, it was decided not to include VIS-L__UV_ images in this study. To reduce further parasitic stray light, image acquisition was done while the surrounding light was switched off. It is worth noting that data acquisition was only possible during the regular opening times of the cathedral, so the image acquisition was slightly affected by surrounding light coming from other parts of the cathedral and a small window in an adjacent niche close to the mural paintings. This might have affected the UV images slightly but not the VIS and IR.

The mural painting was documented in 19 images of UV-R__IR_, VIS-R__VIS_ and two IR-R__IR_ (850 nm and 1000 nm). Eighteen images for capturing the surface (six lines with three images each) and one image with a reflectance target Lastolite EzyBalance 18% grey for the white balance in its centre. In total, 76 images were taken. Hereby, a set of three images (one UV, VIS and IR) was always taken from the same camera position. For each spectral band, the images were acquired using identical camera settings following the CAHRISMA guidelines [10]. For all images, the same focus, ISO 100 and aperture f/13 were used, only the exposure time changed between the individual spectral bands (VIS: 1/3 s; UV: 10 s; IR__850_: 1 s; IR__1000_: 4 s). All images were stored as JPG and RAW files. The JPGs served as a fast overview; all further processing was done by using the RAW files.

#### 2.2.2. Photogrammetric Image Acquisition

The MBI image acquisition was focused on the overall documentation of the mural painting. The images were mostly taken from a frontal position of the camera, being not ideal for photogrammetry [28]. To support the MBI, another image set with a proper photogrammetric image bundle was taken using a Canon EOS 5D Mark II (Table 3). This survey followed a stable photogrammetric workflow, including oblique and tilted images and higher variations of the camera positions (distance and angle left/right around the painting) [28]. The image set contained 17 images, all acquired with identical camera parameters: ISO 100; f/13; and exposure time 0.7–1.5 s.

#### 2.2.3. TLS Data Acquisition

The terrestrial laser scans (TLS) provide ground truth information for referencing the photogrammetric data and give further data about the orientation and localisation of the mural painting in the cathedral. For the TLS survey, a Trimble TX6 was used (Table 4). A single scan position was sufficient to capture the scene of the wall painting, but to provide some additional data on the surrounding area two additional scans were acquired. A standard range-based data acquisition workflow was followed [43]. The TLS internal setup Level 1 was chosen, providing a resolution of 23 mm on 30 m, 34 million points per scan. Since the wall painting was located at a distance less than 5 m from the TLS, the surface resolution of the mural painting increased substantially to approx. <4 mm.

### 2.3. Data Processing

Data processing was separated into several steps. Figure 3 provides an overview of the entire workflow. First was a pre-processing of the raw data in the individual software of each technique. This step was followed by the combination of the different spectral bands and sensor techniques using photogrammetry. From these individual spectral 3D models, it was possible to export orthoimages, forming the basis for the hybrid image analysis and further investigations in GIS. In the following paragraphs, the individual steps of pre-processing, combination and analysis are presented.

#### 2.3.1. Processing TLS data

In the first step, the three TLS scan positions were registered in a local reference system [43]. Processing the data was done in Trimble RealWorks v. 11.3 using a cloud-to-cloud registration. The scans had a high overlapping, thus resulting in an overall registration error of 1.83 mm. In the following step, the X, Y, and Z coordinates of nine natural points located on the mural painting were exported (Figure 4). The points were used as control points (CP) and check points (ChP) for scaling the photogrammetric data. Therefore, the four corners served as CPs and the other points as ChPs.

#### 2.3.2. Pre-Processing Image Data

Before starting the photogrammetric calculation, it was necessary to develop the RAW image files. Processing of the UV-R__UV_, VIS-R__VIS_ and IR-R__IR_ RAW images was done using the free and open-source software RawTherapee v. 5.8. Here, according to the recommendation of the EU project CHARISMA [10] first, the tone curve was set to linear to eliminate automated, camera-specific colour manipulations, and the white balance was adjusted based on the reflectance target. The IR-R__IR_ and UV-R__UV_ images were converted to monochrome images, as they represent spectral ranges not visible to the human eye. Therefore, the UV-R__UV_ was reduced to its blue channel and the IR-R__IR_ images to their red band. Since all images of the same spectral range were taken under identical conditions (surrounding light, artificial light source and camera settings) the image settings were calculated on the RAW images containing the reflectance target and then transferred to the other images of the spectral batch. All images were exported as uncompressed TIFFs [23]. The supporting colour images of the Canon EOS 5D Mark II were also processed using RawTherapee, according to the workflow mentioned above, and exported as TIFFs.

Working with uncompressed TIFFs is very important for image analysis, especially for processing the principal component analysis (PCA) and the image calculations for the different false colour and hybrid images. Using lossy compression images such as JPGs may cause artefacts [44], compromising the final results and thus possibly causing misinterpretation.

#### 2.3.3. Photogrammetric Data Processing

The software Agisoft Metashape v. 1.7.4 was used for photogrammetric processing and orientation of the MBI and colour images. In the first step, the CPs and ChPs exported from the TLS data and all spectral and colour image files were imported. A standard photogrammetric SfM-based workflow was followed [28]:Internal camera orientation (all images: MBI and colour);Relative external orientation and tie point extraction (all images: MBI and colour);Absolute external camera orientation (all images: MBI and colour), using CPs and evaluation using ChPs; root mean square error (RMSE) is shown in Table 5;Generation of a dense point cloud (only colour images);Generation of a 3D model (only colour images);Texturisation of the 3D model (separate for each spectral band: UV-R__UV_, VIS-R__VIS_, 2x IR-R__IR_). To avoid image compression during texturisation, a TIFF-based texture mapping was chosen.

For further image analysis, four high-resolution orthoimages were produced, one for each spectral band (UV, VIS, IR 850 nm and IR 1000 nm; Figure 5). The orthophotos had a GSD of 0.3 mm and identical image sizes. They were exported as uncompressed GeoTIFFs containing the image file in TIFF format, and an additional world file, TFW, to provide the orientation and scaling of the image. The world file has the same file name as the TIFF and is based on a simple text file with six lines, providing the following information:1st line: pixel size in metres for the X-axis;2nd line: image rotation around the Y-axis;3rd line: image rotation around the X-axis;4th line: pixel size in metres for the Y-axis (negative value);5th line: X-coordinate for the upper left pixel;6th line: Y-coordinate of the upper left pixel.

With this information, it is possible to import the orthoimages perfectly aligned and scaled into GIS.

## 3. Results

Based on the MBI orthoimages, a wide range of image analyses can be executed for further investigation of the mural painting. Hereby, the approaches presented in the following paragraphs lead to new image files based on the interaction of two or more spectral images. These resulting artificial images are called hybrid images or hybrid orthoimages.

### 3.1. Hybrid Image Processing and Image Analysis

The creation of hybrid images by mixing and combining different bands sometimes provides the opportunity to detect, highlight and visualise material differences better than the single monochrome image band of a single spectral range [9]. These hybrid images may follow the well-known use of UV and IR false colour images, but also other combinations and arithmetic calculations, e.g., principal component analysis (PCA) and the normalized difference vegetation index (NDVI) can reveal important information. The different image-processing approaches are discussed in the following paragraphs. Since all resulting hybrid images are based on the multiband orthoimages, they have identical pixel numbers and sizes. Through this, it is easily possible to copy a world file of one of the multiband orthoimages by changing its file name according to the specific hybrid image. This will secure the exact superimposition of all multiband and hybrid images later in GIS.

#### 3.1.1. False Colour Images

False colour (FC) images are well known in the multiband image analysis of CH [9,10,14,15,23,29]. Hereby three different monochrome spectral bands are combined into one R/G/B image, resulting in a hybrid image containing the information of the three bands. The most common false colour images are UV false colour (FCUV) and IR false colour (FCIR). For FCUV, the monochrome UV-R__UV_ image is combined with the green (~545 nm) and blue (~460 nm) channels of the VIS-R__VIS_ image to create a new image. In this hybrid image file, the R/G/B channel corresponds to the G/B/UV band of the initial data. For the FCIR, the monochrome IR-R__IR_850_ image is combined with the red (~645 nm) and green channel of the VIS-R__VIS_ to form a new image file (Figure 6). In the FCIR, the R/G/B channels of the hybrid image correspond to the IR/R/G bands of the initial monochrome spectral images. FCIR and FCUV images are mostly used for the classification and differentiation of materials, e.g., inks, dyes, and pigments [9,20,23,45,46,47,48].

In the case of the ‘Adoration of the Shepherds’, FCIR helps to compare the blue pigments used for the sky and Marias veil (Figure 6). Both blue areas behave identically in the FCIR and turn into a dark reddish colour. Assuming this, probably the same blue pigment was used by the artist to draw both areas. Turning the blue into a strong red in the FCIR may indicate the presence of lapis lazuli as the main blue pigment used here [47]. Further analysis is mandatory to clarify this suggestion, even if some tests carried out with VIS-L__UV_ in the blue areas also suggest the idea of the use of lapis lazuli by showing a luminescence that could easily fit the one reported for such a pigment [15]. For sure, it does not seem to be azurite, which turns into dark purple in FCIR, but we cannot underestimate smalt (light red in FCIR) or even mixtures of pigments [18]. Anyway, the natural ultramarine pigment was requested as the main blue to be used by these painters in the angels ceiling paintings, as stated in the clauses of the contract [40], so it seems logical that it would have also been used here if they were going to show their ability with such pigment.

#### 3.1.2. Principal Component Analysis (PCA)

PCA is a method for the statistical analysis of data. Applied to image data, it enables an unsupervised extraction of qualitative information of compounds. For example, a standard RGB colour photo can serve as the data basis, where the three colour channels represent different spectral bands (red, green and blue) and are superimposed with pixel accuracy. Varying material characteristics in the different spectral ranges can be compared and analysed pixel by pixel using PCA. Hereby, Principal component 1 (PC1) shows the matrix of the maximum variance between the initial data. The values calculated in this way can then be visualised in a greyscale image. Principal component 2 (PC2) corresponds to a differentiated matrix, where PC1 has been subtracted from the original matrix. In this continuation, any number of PCs can be calculated. Details and differences in the material composition can be filtered and represented in this way in continuous PCs [49,50]. The advantage of PCA is the comparatively simple evaluation of an unlimited number of spectral bands, as long the individual image bands are correctly superimposed [51,52].

Calculation of the PCA of multiband orthoimages was done using the freeware HyperCube v. 11.52. Therefore, the orthoimages were imported and superimposed in an image cube. The VIS orthoimage provides three bands (red, green and blue) and the other monochrome spectral orthoimages, each one band. Using identical settings for processing the orthoimages guaranteed a pixel-accuracy superimposition of the single bands. The principal components (PC) were exported as individual TIFF files (Figure 7). Since PCA is processed on the multiband orthoimages, the single PCs have identical pixel sizes. In this way, it is easily possible to create world files for each PC and hybrid image by reusing a world file of one of the initial orthoimages to secure the exact positioning later in GIS.

For better feature extraction and observation of the PCs, the combination of the monochrome images into hybrid false colour images provides advantages [52]. For the investigation of the ‘Adoration of the Shepherds’, two hybrid images based on PCA are suitable. First is the combination of PC 1, 2 and 3, shown in Figure 8. This hybrid orthoimage helps to understand the unfinished areas in the lower area of the painting. The underdrawings, for example, of the contour of Marias veil, the hands of the left shepherd and the halo of the undrawn Jesus are much more visible than in the VIS-R_VIS (Figure 1) or the monochrome multiband images (Figure 5). This helps a lot in understanding the composition of the painting. The location of the newborn baby Jesus, especially, in the unfinished areas of the painting is very difficult to identify without PCA and hybrid images. Besides the figural concept, the combination of PC 1, 2 and 3 helps to distinguish between different preparation layers by increasing the contrast of the border lines of the plaster and the surface roughness.

The second promising hybrid orthoimage is based on the combination of PC 3, 4 and 5, shown in Figure 9. This image provides several important pieces of information. First is the combination of surface details only visible in the UV-R__UV_ and covered pentimenti only visible in the IR-R__IR_ in one single image. This provides information on two hardly visible attributes. In this image, the pentimenti of a shepherd’s crook in the background behind the right shepherd becomes visible (Figure 9c). Furthermore, Paolo de San Leocadio and Francesco Pagano equipped Joseph with another rod too. In the landscape behind Joseph, the top of a twisted rod, drawn in a secco technique on top of the fresco of the landscape, is visible but only hard to identify in the VIS-R__VIS_ (Figure 1). The hybrid image reveals the extension of the rod as an underdrawing in front of Joseph’s lap (Figure 9b).

Furthermore, the hybrid image is suitable to detect material differences and damages in the mural painting. Figure 9d shows details of the sky, revealing clouds but also two different damage phenomena. Both damage phenomena may be caused by a loss of the surface. The greenish damages represent areas with a loss of the paint layer and uncovered plaster surface. They have the same optical behaviour as the bare plaster in the lower part of the painting. The blackish areas also represent areas with a loss of the surface, but they are almost invisible in the VIS-R__VIS_ (Figure 9e). It seems these areas are covered by another paint layer using different pigments or binders, applied during a former restoration.

#### 3.1.3. NDVI/NDPI

In addition to the PCA and superimposition of single bands in hybrid images, several tests by applying the normalized difference vegetation index (NDVI) algorithm to the image files were carried out. NDVI is normally used in landscape observation to detect changes in vegetation and its condition. The formula is based on comparing the near infrared band and the visible red band, based on the fact that healthy green vegetation reflects more NIR and less VIS, while sparse and weak less green vegetation reflects more VIS and less NIR [53]:NDVI = (NIR − RED)/(NIR + RED) 

In this study, the formula was adopted to process a hybrid image of the UV and IR spectral bands. Therefore, the formula was adjusted to the monochrome UV-R__UV_ and the IR-R__IR_ images to extract and visualise their differences, yielding a new index that we have called NDPI, ideal for paintings instead of vegetation:NDPI = (UV − IR)/(UV + IR) 

The result is shown in Figure 10. In the case of the mural painting, the advantage of this hybrid image is better detection of surface damages, e.g., in the face of the right shepherd appear stronger contrasts between areas with a loss of the surface and areas with intact paint layers (Figure 10c). Furthermore, the surface roughness is highlighted, thus simplifying the detection of plaster borders of adjacent *giornata* (Figure 10b).

### 3.2. Combination in GIS

After processing the hybrid images, the combination of all resulting image data was necessary for a better comparison and analysis of the mural painting. This was done in a geographic information system (GIS), by using the open-source software QGIS v. 3.26.3. As already mentioned, all hybrid images were created by using the multiband orthoimages and saved in TIFF format. This means that all resulting image files had identical pixel numbers, size and orientation to the initial multiband orthoimages, so, because of this, it was easily possible to create world files for each hybrid image by copying the world file of one of the initial multiband images. When all image files are equipped with a world file, they can be imported as single layers into GIS, perfectly superimposed to each other (Figure 11).

Working in a GIS environment provides several advantages. The software can handle both raster data (i.e., image files) and vector data. In this way, it is possible to adjust the colour ramp of the monochrome image files (MBI and PCA image files) to enhance the contrast or extract further information for the analysis. However, it is also possible to map findings and features by tracing with vectors. By simple switching between the image files, a comparison is very easy and the tracings can be overlaid on every image file. By working with exactly scaled orthoimages, it is possible to extract metrical correct plan sets for the dissemination and archiving of the results. Figure 12 shows a final plan of the VIS-R__VIS_ multiband image with overlaid tracings of the *giornata*, the underdrawings and the *pentimenti* extracted from the different hybrid images.

## 4. Discussion

In the previous chapters, a workflow for the analysis and investigation of wall paintings by using multiband images and hybrid image analysis was presented. Hereby, a workflow for the photogrammetric combination of different MBI images was chosen, aiming to capture even larger wall paintings in a high spatial resolution.

The flexible workflow avoiding complex setups (e.g., following rails or a strict raster) allows moving the camera on a simple tripod. This makes it easy to handle and adjustable to the surrounding conditions and the needs of the heritage object. Following this workflow, even objects with complex surfaces, e.g., curved ceilings or natural caves, can be properly captured.

In this paper, referencing of the photogrammetric data was carried out by using TLS data and natural CPs and ChPs. This allowed the inclusion of the data in an overall data set of the entire heritage object, providing the possibility to cross-compare the data with MBI data from other areas, but, most importantly, was the possibility to allow an overlying of the images with MBI data from additional campaigns for monitoring [54].

As shown in Table 5, the RMSE of the CPs and ChPs was 8 and 14 mm, being very high values, whereas the pixel error was 1.9 and 3.4 pixels. This error was caused by picking natural target points on the TLS point cloud of the mural painting manually. Hereby, the point cloud does not have enough contrast and resolution for a more accurate selection of the targets compared with a more accurate selection in the images. Next time, scaling and referencing will be separated to avoid this problem. For scaling, scale bars with coded targets will be included in the image scene while capturing the MBI and photogrammetric images to guarantee a correct submillimetre scaling, and for referencing better identifiable points such as artificial targets. Alternatively, a best-fit matching of the MBI point cloud to the TLS point cloud can be processed. Hereby the entire geometry of the MBI data can be used for referencing.

As shown in the results, standalone MBI is a low-cost, straightforward but complex method for the non-destructive investigation of mural paintings. The single spectral bands provide much information about the materials and condition of a painting. However, to extract all the information, a manual comparison of the single bands is difficult and time-consuming. Therefore, processing hybrid images to extract and differentiate all this plethora of multiband information processing is considered a powerful tool. The well-known and established combination of the spectral bands to IR and UV false-colour images already provides hints for pigment identification and differentiation (Figure 6), but the image files contain much more information.

Processing hybrid images based on PCA is identified especially as a suitable method to reveal easily complex multiband information. The PCA false colour images can combine information provided by all spectral bands, making it possible to trace features from the UV up to IR in one image, as shown on the shepherd’s crook (Figure 9). Coatings or interventions from earlier conservation measures can also be identified with this method (Figure 9c). Such information is normally provided in VIS-L__UV_ images, which presupposes the full absence of surrounding light and properly filtered UV-light [23]. These requirements were not given during image acquisition. Therefore, a second documentation campaign under ideal conditions will be carried out in the near future to compare the benefits of integrating the VIS-L__UV_ into hybrid image processing.

It is worth mentioning that the hybrid images based on PCA provide very colourful pop-art-like images. These results may confuse the heritage expert and it might require additional unexpected time to feel comfortable working with them. Nevertheless, the outstanding results are highly recommended as good practice to get the best from on-site surveys, very difficult to achieve with conventional limited resources, e.g., conventional cameras and visible light. PCA allows the identification of even the finest faint details [52].

NDVI has long been used to characterise vegetation in the literature [55,56]. A simple variant of it, herein called NDPI adapted for the investigations of paintings, shows some good results in differentiating surfaces or material differences with similar colour behaviour in VIS-R__VIS_ and for the detection of *giornata* (Figure 10). Since it was the very first time adopting NDPI for mural paintings, further applications and tests on other objects are mandatory to confirm its functionality and benefit in a wider context.

## 5. Conclusions

In this paper, the investigation of MBI images of a mural painting is presented. Hereby a photogrammetric workflow to capture multiple spectral bands of large areas in high resolution is followed. To analyse the MBI image set, new processing workflows to deal with hybrid images based on PCA and NDPI are shown.

Hereby, the hybrid images based on PCA allow users to reveal features and detect material differences and damages hardly visible in the image of a single spectral band. This workflow simplifies the time-consuming observation for feature extraction of the single MBI images and helps to investigate an overall overview of all spectral bands involved.

NDPI is a new method for the image-based analysis of paintings based on NDVI. In the case study under investigation in this paper, NDPI helps, for example, to identify *giornata* and material differences in areas of low colour contrast.

All hybrid images are combined and superimposed with the MBI image set in a sub-pixel accuracy in GIS, making it possible to switch between the single image files for comparison and analysis of all features and details. Furthermore, metrically correct plans are easily exportable for dissemination and archiving from the GIS.

Analysing the hybrid images reveals several important pieces of information for the case study, the ‘Adoration of the Shepherds’. The FCIR image indicates strong similarities between the blue pigments of Marias veil and the sky, whereby they turn into a deep red in the FCIR, indicating the presence of lapis lazuli as the main pigment used. Further checks, e.g., its behaviour in VIS-L__UV_ or a sample analysis, may verify this suggestion.

Furthermore, the hybrid images help to visualise the underdrawings in the lower part of the unfinished painting better. In combination with the traces of the *giornata* detected with NDPI, the results allow experts to achieve a better understanding of the artists’ image composition workflow.

## Figures and Tables

**Figure 1 sensors-23-02301-f001:**
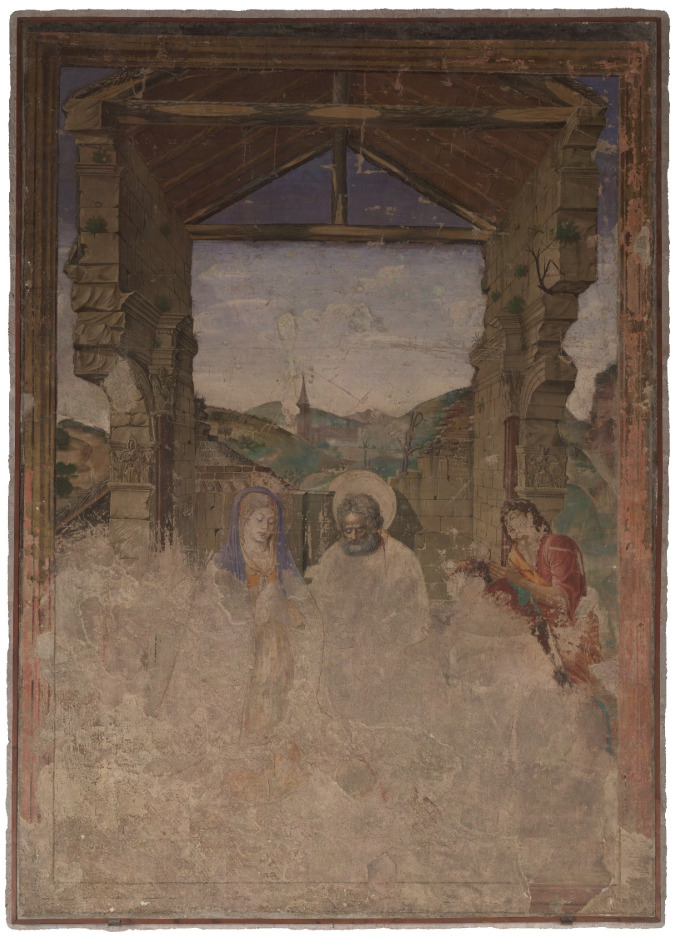
Overview of the “Adoration of the Shepherds” by Paolo de San Leocadio and Francesco Pagano (1472).

**Figure 2 sensors-23-02301-f002:**
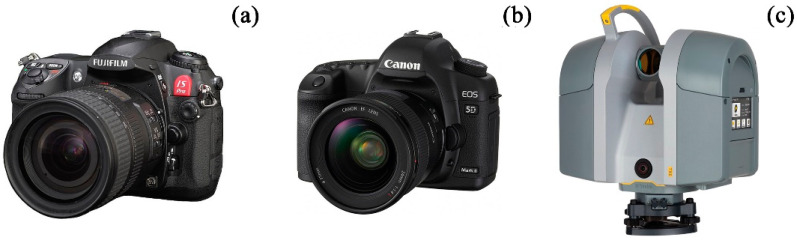
(**a**) Fuji IS Pro; (**b**) Canon EOS 5D Mark II; (**c**) Trimble TX6.

**Figure 3 sensors-23-02301-f003:**
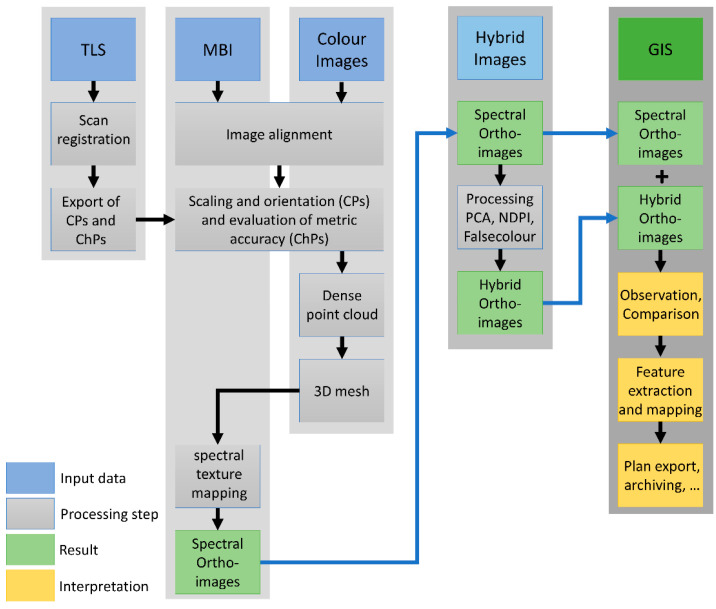
Schematic overview of the workflow for data processing and data fusion.

**Figure 4 sensors-23-02301-f004:**
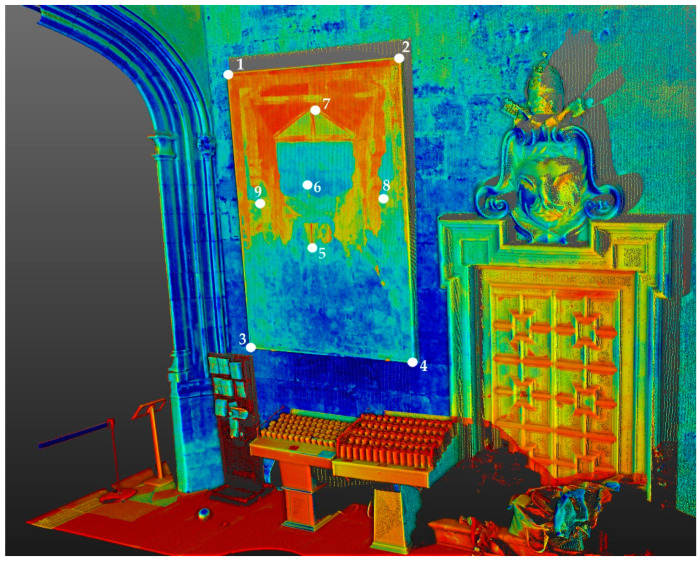
Reflectance image of the TLS point cloud, showing the mural painting and its surrounding situation in the Cathedral of Valencia. The CPs (1–4) and ChPs (5–9) are highlighted in white.

**Figure 5 sensors-23-02301-f005:**
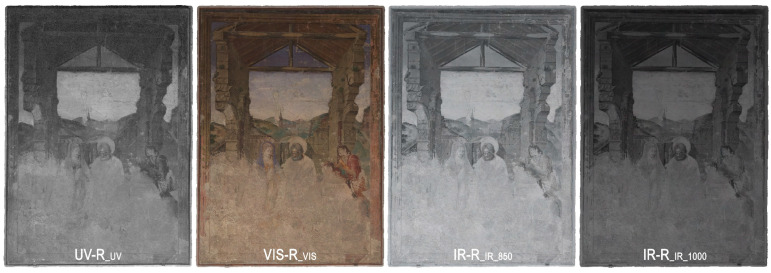
Overview of the MBI orthoimages.

**Figure 6 sensors-23-02301-f006:**
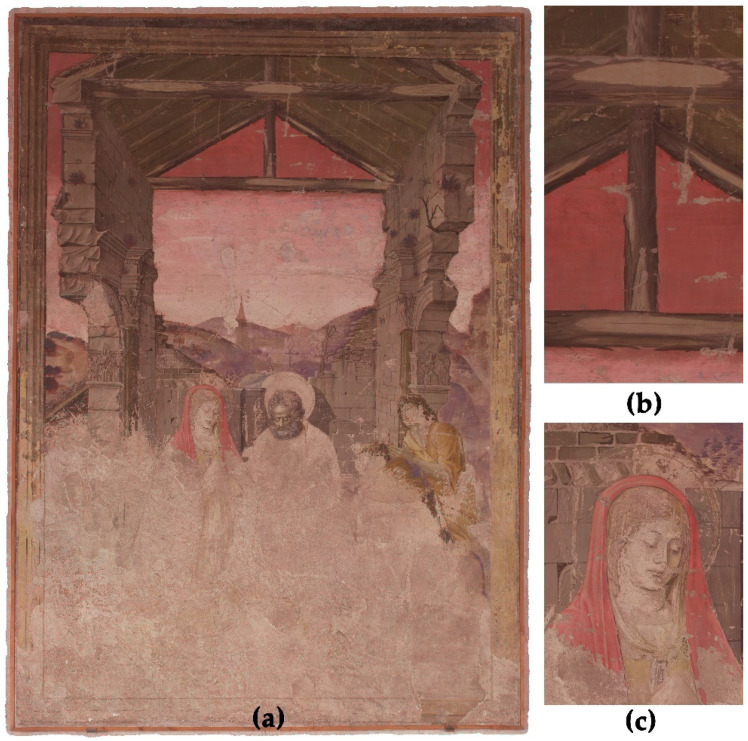
(**a**) False colour infrared image (FCIR) based on the monochrome spectral bands of IR-R__IR_850_ and the red and green bands of the VIS-R__VIS_. This hybrid image helps to compare the blue pigments used in the mural painting. The blue pigments used for the sky (**b**) and for Marias vail (**c**) behave identically in FCIR.

**Figure 7 sensors-23-02301-f007:**
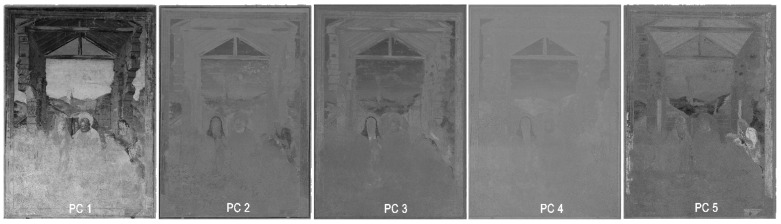
Overview of the PCA orthoimages.

**Figure 8 sensors-23-02301-f008:**
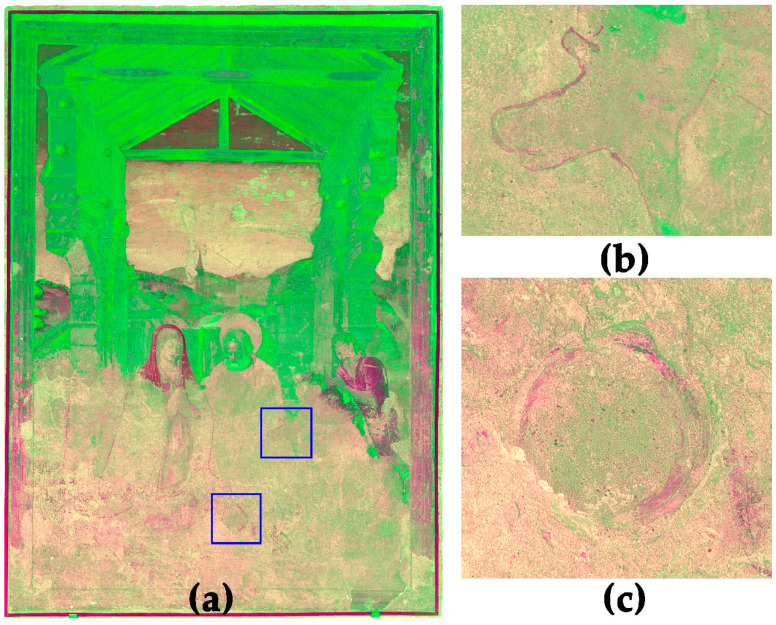
(**a**) Hybrid image based on the combination of PC1, 2 and 3. The image helps to localise underdrawings in the unfinished lower part of the mural; (**b**) detail of the shepherd’s hand; (**c**) contour line of the halo of the newborn baby Jesus.

**Figure 9 sensors-23-02301-f009:**
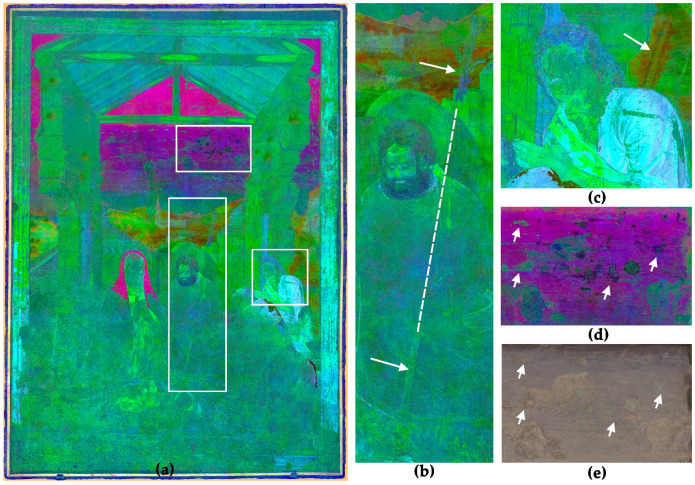
(**a**) Hybrid image based on the combination of PC3, 4 and 5. The image helps to localise covered underdrawings and material differences (highlighted with arrows): (**b**) features and tracing of a rod belonging to Joseph; (**c**) *pentimenti* of a not-realised shepherd’s crook; (**d**) surface damages and restorations with different materials; (**e**) same detail in VIS-R__VIS_.

**Figure 10 sensors-23-02301-f010:**
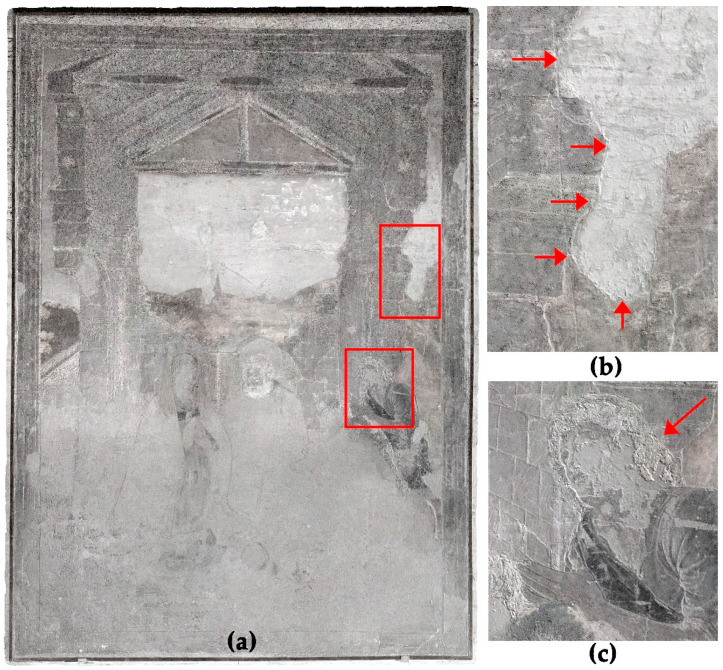
(**a**) NDPI image based on the UV-R__UV_ and the IR-R__IR_ spectral images. (**b**) Detail showing the plaster borders between two *giornata*. (**c**) Detail highlighting the differentiation of intact and lost surfaces.

**Figure 11 sensors-23-02301-f011:**
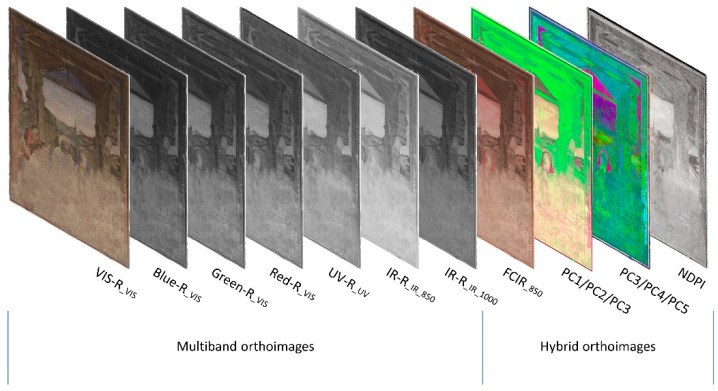
Multiband orthoimages and hybrid orthoimages superimposed in GIS.

**Figure 12 sensors-23-02301-f012:**
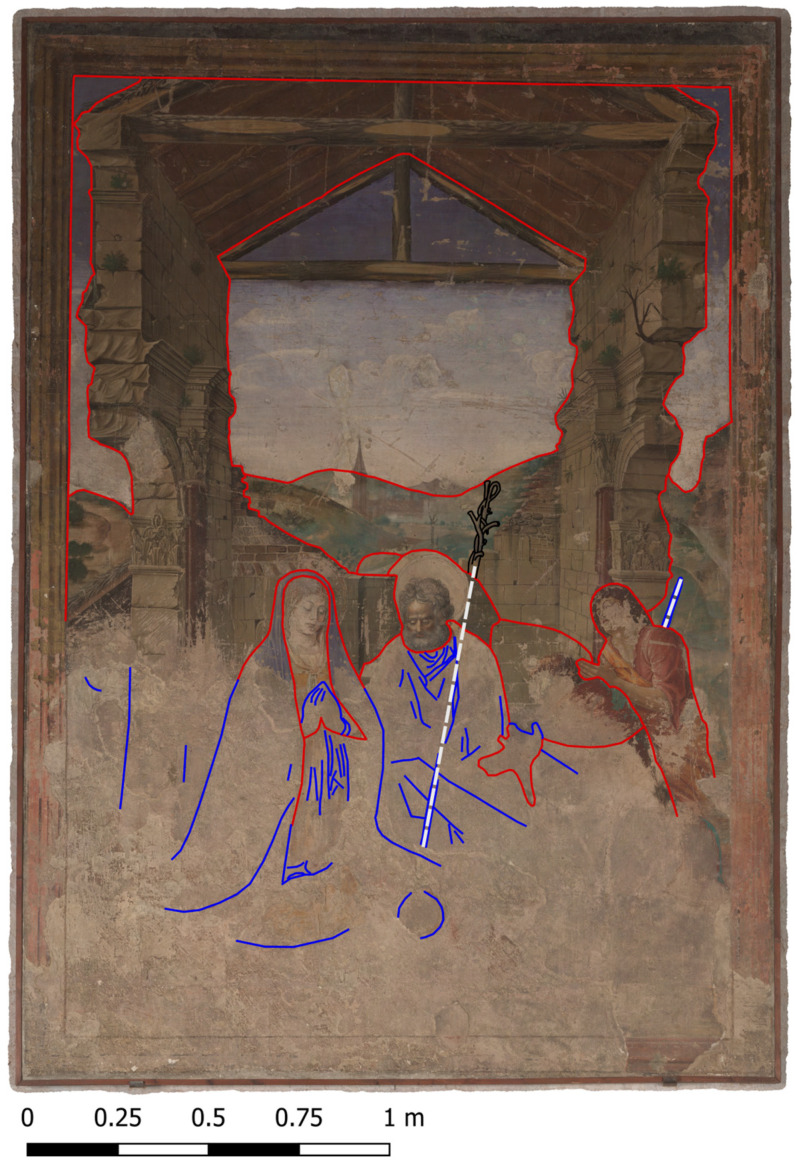
Visible induced visible reflectography image with tracings based on the hybrid images. Red: plaster borders, *giornata*; Blue: Underdrawings and *pentimenti*; White: reconstructed line of the shepherd’s crooks; black: highlighted top of Joseph’s rod.

**Table 1 sensors-23-02301-t001:** Overview of spectral images commonly used in MBI.

Abbreviation	Imaging Method	Also Known as	Radiation Source	Filter Sensitivity
UV-R__UV_	UV-induced UV reflectography	UVR, UV-reflectance imaging	UV (~365 nm), UV-LED, Wood’s lamp	UV band-pass filter (320–390 nm)
VIS-L__UV_	UV-induced VIS luminescence imaging	UVL, UVF (UV-fluorescence) imaging	UV (~365 nm), UV-LED, Wood’s lamp	UV/IR-cut filter (380–750 nm)
VIS-R__VIS_	VIS-induced VIS reflectography	VIS, VIS-reflectance imaging, Colour photography	LED, Tungsten, Flash	UV/IR-cut filter (380–750 nm)
IR-L__VIS_	VIS-induced IR luminescence imaging	VIL, IRF (IR-fluorescence) imaging	VIS 380–750 nm	IR long-pass filter (e.g., starting at 850 nm or 1000 nm)
IR-R__IR_	IR-induced IR reflectography	IR, IRR (IR-reflectography) imaging	IR (700–1050 nm), LED or undefined light sources with a high amount of IR radiation, e.g., Tungsten, Flash and Halogen lamps	IR long-pass filter (e.g., starting at 850 nm or 1000 nm)

**Table 2 sensors-23-02301-t002:** Main specifications of the equipment used for MBI data acquisition.

Fujifilm IS Pro		Filters			Light Sources	
Sensor	Super CCD Pro; 12 Mpix.	UV-R	MidOpt BP365 + MidOpt SP730	320–410 nm	2 UV LEDs	UV-R
Sensor size	APS-C; 23 × 15.5 mm	VIS-R	MidOpt BP550	405–690 nm	2 Halogen spotlights	VIS-R + IR-R
Image size	4256 × 2848 pixels	IR-R__850_	MidOpt LP850	850–1100 nm		
Lens	CostalOpt. UV-VIS-IR 60 mm Apo Macro	IR-R__1000_	MidOpt LP1000	990–1100 nm		
Focal length	60 mm					

**Table 3 sensors-23-02301-t003:** Specifications of the DSLR camera used for the photogrammetric survey.

Canon EOS 5D Mark II	
Sensor	CMOS 21.1 Mpx
Sensor size	Full frame (36 × 24 mm)
Image size	5616 × 3744 pixels
Focal length	24 mm

**Table 4 sensors-23-02301-t004:** Specifications of the Trimble TX6 terrestrial laser scanner.

Trimble TX6 Laser Scanner	
Distance measurements	Phase shift
Wavelength	1.5 µm
Extended range	120 m
Horizontal and vertical range	360°/317°
Distance accuracy	<2 mm (1 sigma)
Acquisition speed	500,000 pts/s
Camera	RGB

**Table 5 sensors-23-02301-t005:** RMSE of the CPs and ChPs for the orientation of all image files.

RMSE (m)					
	X (m)	Y (m)	Z (m)	XYZ (m)	Pixel Error
CPs (4)	0.0042	0.0056	0.0039	0.0080	1.902
ChPs (5)	0.0076	0.0072	0.0097	0.0143	3.421

## Data Availability

Requests for access to the data presented in the study should be made to corresponding author.

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
