# Peer review of "Multiband Photogrammetry and Hybrid Image Analysis for the Investigation of a Wall Painting by Paolo de San Leocadio and Francesco Pagano in the Cathedral of Valencia"

_sensors, 2023, doi:10.3390/s23042301_

Round 1
Reviewer 1 Report
The paper contributes to the production of extra new knowledge for the non destructive methodology for studying mural paintings. The paper is well structured. The objectives, the goals and the results are clearly presented. The discussion and the conclusions are supported by the results. However, there are some minor revisions that the authors could take into account. In particular:
l.240: please specify what kind of reflectance target is.
L255: please specify the variations of the camera positions
L366-367, L369: please specify the wavelength of green, blue and red channels
L377-381: please specify the FCIR and FCUV colors for the pigments mentioned and add the relative references for this.
Author Response
Thank you very much for your review. We considered all your suggestions to improve the paper.

Reviewer 2 Report
- This study is very well documented and structured, presenting new workflow used for investigation of an important mural painting. So, flexible approach to capturing and analysing high-resolution photogrammetric MBI data of wall paintings is presented and the application of hybrid images by combining different spectral bands into false colour images is presented, mentioning the advantages of PCA and NDPI method for the investigation of mural paintings. The authors follow the teminology of Keller about Multiband Imaging, most appropiate for cultural heritage imagistic investigations.
- The novelty of this study referee to the results obtaining by different spectral imaging techniques was the data acquisition and data processing, image analysis, hybrid image processing and especially further investigation in a Geographic Information System (GIS).
- For cultural heritage combination of MBI, a simple and non distructive method for investigation of mural paintings with other modern methods (in this case NDPI, GIS) is very important for understanding old materials and technics.The authors follow the teminology of Keller concerning Multiband Imaging.
2 little observations:
- R 228 mainly light around 365 nm not manly
- - In Figure 4 CP and ChP are too small written
Author Response

(The authors gave the same response as above.)
